# Phytochemical Composition and Antioxidant Activity of Pot Marigold (*Calendula officinalis*): The Impact of Light Modification

**DOI:** 10.3390/plants14223512

**Published:** 2025-11-18

**Authors:** Lidija Milenković, Zoran S. Ilić, Ljiljana Stanojević, Aleksandra Milenković, Ljubomir Šunić, Bratislav Ćirković, Dragan Božović, Žarko Kevrešan

**Affiliations:** 1Faculty of Agriculture, University of Priština in Kosovska Mitrovica, 38219 Lešak, Serbiabratislav.cirkovic@pr.ac.rs (B.Ć.); 2Faculty of Technology, University of Niš, Bulevar Oslobodenja 124, 16000 Leskovac, Serbiaaleksandra.milenkovic@student.ni.ac.rs (A.M.); 3Research and Development Institute-Tamiš, Novoseljanski Put 33, 26000 Pančevo, Serbia; 4Institute of Food Technology, University of Novi Sad, Bulevar Cara Lazara 1, 21000 Novi Sad, Serbia

**Keywords:** pot marigold, edible flower, phenols, flavonoids, antioxidant activity

## Abstract

This study evaluated the impact of colored shade nets (control, pearl, red, and blue) on the phytochemical composition and antioxidant activity of two pot marigold (*Calendula officinalis* L.) varieties: the ‘Springer mix’ from Italy and a ‘domestic’ variety from Serbia. Phytochemical parameters assessed included total extractive matter (TEM), total phenolic content (TPC), total flavonoid content (TFC), and antioxidant activity measured by DPPH and FRAP assays. The results indicate that colored shade nets significantly enhanced the accumulation of bioactive compounds and antioxidant activity, particularly in the ‘domestic’ Serbian variety, which consistently outperformed the Italian ‘Springer mix’. Blue nets notably improved extract yield and radical scavenging activity, red nets promoted flavonoid synthesis, while control conditions led to the highest TPC. The ‘domestic’ Serbian variety exhibited exceptionally high TPC (68.21 mg GAE/g dry extract) and TFC (55.55 mg RE/g dry extract) content. Correspondingly, potent antioxidant activity was observed, with EC_50_ values as low as 0.065 mg/mL under blue net conditions. Principal component analysis further validated the superior phytochemical profile of the Serbian variety. These findings demonstrate the effectiveness of light spectrum manipulation in enhancing the medicinal quality of pot marigold and underscore the potential of Serbian germplasm for high-value cultivation in phytopharmaceutical applications.

## 1. Introduction

The genus *Calendula* L. (family Asteraceae) comprises approximately 25 species, among which *Calendula officinalis* L. and *Calendula arvensis* L. are the most prominent. These species are predominantly distributed across the Mediterranean region. *Calendula officinalis* L. commonly known as pot marigold, is an annual or short-lived perennial herbaceous plant with multiple applications. In organic gardens pot marigold has beneficial effects: its root secretions repel soil-dwelling pests, particularly nematodes, and protecting neighboring vegetables (white cabbage, from aphid parasitoid-*Diaeretiellarapae*) [1]. It is widely utilized as an ornamental plant, a component in cosmetic formulations, a medicinal species with various therapeutic properties, and as an edible garnish in culinary practices [2,3]. Pot marigold exhibits a broad spectrum of medicinal properties. Traditionally, its leaves have been used as diaphoretic and expectorant agents, while the flowers are recognized for their antispasmodic effects and are employed to stimulate menstrual flow. In Serbian traditional medicine, pot marigold flowers are widely utilized in the preparation of cream bases and various unclassified topical formulations for the treatment of skin infections, wounds, burns, and rashes [4].

Pot marigold suspension or tincture is used to treat acne locally, reduce inflammataion, control bleeding and soothe irritated tissue. Pot marigold plant’s extract, have been show to have anti-inflammatory, antioedematous, antioxidant activity, anti-bacterial and antifungal activity, antiviral activity, wound healing and immunostimulant activity [5]. In traditional French medicine, pot marigold flowers are commonly prepared as a herbal infusion to alleviate fever and induce perspiration [6]. Based on ethnobotanical knowledge from eastern Serbia, the pot marigold flower is most commonly used in ethnomedicine [7,8], while in other regions of Serbia, the flowers are traditionally used as a spice in soups, broths, and various dishes, andthe leaves incorporated into salads [9]. As edible flowers pot marigold species are known for their antioxidant potential [10]. The use of edible flowers as a complex source of antioxidant compounds (carotenoids, terpenoids, phenolic acids (*p*-coumaric, caffeic, ferulic, and ellagic), flavonoids (procyanidin A_2_ and quercetin-3-rutinoside) can be a good alternative to replacing synthetic food additives in health-promoting applications. Edible flowers are increasingly becoming an integral part of functional foods as health-beneficial additives, and marigold, with its phytochemical components, is gaining more presence. In their study of different marigold genotypes, Sultana et al. [11] report that pot marigold is rich in bioactive compounds -total phenolic (TPC) (428.58 to 592.71 mg gallic acid equivalent (GAE)/100 g) and flavonoid content (TFC) (135.06 to 233.39 mg quercetin equivalent (QE)/100 g). Bragueto Escher et al. [12] studied the chemical composition and antioxidant capacity of pot marigold flower and incorporated the extract of marigold into an organic yogurt. Pot marigold flower extract is used in soothing cosmetics, such as after-sun, for sensitive skin and eye contour products [13]. Pot marigold can be cultivated both in controlled environmental conditions (such as greenhouses, vertical farming in plant factory with artificial lighting) and in open fields. With respect to light requirements, pot marigold is classified as a facultative long-day plant, meaning it can flower under any photoperiod. Such conditions allow for better organization of production throughout the entire year. Light conditions, in particular, play a crucial role in determining not only yield but also plant quality. The most important light-related parameters are photoperiod, light intensity, and light spectrum [1]. The application of photoselective shade nets provide the plants with more optimal growth conditions [14]. For energy-efficient production, it is advisable to cultivate pot marigold under low light conditions, with a recommended intensity of around 200 µmol·m^−2^·s^−1^ (PPFD) [15]. In open-field conditions, light environments can be partially modified by using colored photo-selective shading nets to better adapt to the needs of pot marigold plants. Based on previous results, the use of colored nets can positively influence both the yield and quality of a wide range of medicinal and aromatic plants; sage [16] coriander [17] and basil [18]. Light is recognized as a key ecological factor influencing plant morphology, physiology, and the accumulation of bioactive compounds. In pot marigold light conditions may significantly affect both quality parameters and the biochemical composition of the flowers.

Based on this premise, the present study was designed to test the hypothesis that modified light conditions can alter the accumulation of secondary metabolites. Specifically, the aim was to evaluate the impact of shading with colored nets (pearl, red, and blue) compared to unshaded open-field cultivation on the content of total extractives, phenolic compounds, flavonoids, and antioxidant of pot marigold flowers. These studies will contribute to the development of new techniques aimed at creating optimal microclimatic conditions to increase yield, enhance the content of active compounds, and improve the efficacy quality of marigold.

## 2. Results

Color shading nets combined with physical protection (from birds, insects, hail, wind, and disease carriers) help to reduce light intensity while also modifying its spectral composition. This creates a microclimate that supports plant metabolism and enhances physiological processes influenced by light. For example, during a sunny July day in an open field without shading, the maximum recorded solar radiation was 874 W/m^2^. When using colored shading nets, however, there was a noticeable reduction in net radiation. The most significant decrease in radiation was observed with the blue nets, which recorded 459 W/m^2^.

Additionally, pearl nets with 40% shading were found to reduce photosynthetically active radiation (PAR) more intensely than the other nets or the control (open field). The PAR in the shaded condition was measured at 1100 µmol m^2^ s^−1^, compared to 2242 µmol m^2^ s^−1^ in the open field.

Light quality in terms of color and wave length can affect the morphological structure of the plants. Therefore, the change in light quality and different spectra passing through the nets in controlled environments may improve the quantity and quality of the plants. Red and far-red light can increase stem length and accelerate flowering. The pot marigold plants grown under red shading nets produced the highest plant height, number of flowers, with the largest flower diameter and the total yield compared to the other nets and the open-field plants (Table 1).

Overall, it was found that the application of photoselective shade nets provided the plants with more optimal growth conditions than when they were not used. It seems that the passing of photosynthetically active radiation from the red shade net, i.e., red and orange spectra of visible light, enhanced important plant phenomena, such as photosynthesis and consequently, the biomass.

The effects of colored cover nets (control, pearl, red, and blue) on the phytochemical composition and antioxidant activity of two pot marigoldvarieties (‘Springer mix’ from Italy and ‘domestic’ from Serbia) were evaluated. The results, including total extractive matter (TEM), TPC, TFC, and antioxidant activity (DPPH and FRAP assays), are presented below, with significant differences determined by one-way ANOVA followed by Duncan’s multiple range test (*p* < 0.05).

### 2.1. Total Extractive Matter (TEM)

The TEM, expressed as grams per 100 g of dry plant material (g/100 g d.m.), varied significantly across treatments and varieties (Table 2). The Springer mix variety under blue nets exhibited the highest TEM (31.23 ± 0.237 g/100 g d.m.), significantly higher than all other treatments (*p* < 0.05). The domestic Serbian variety under blue nets also showed a high yield (30.90 ± 0.118 g/100 g d.m.), closely followed by the control treatment of the same variety (30.02 ± 0.227 g/100 g d.m.). The lowest TEM was observed in the ‘Springer mix’ variety under pearl nets (22.29 ± 0.381 g/100 g d.m.), indicating that pearl nets may reduce extract yield in this variety. The ‘domestic’ Serbian variety generally outperformed the ‘Springer mix’ in control and pearl treatments, suggesting varietal differences in extractable matter accumulation.

### 2.2. Total Phenolic Content

Total phenolic content (TPC), expressed as mg gallic acid equivalents per gram of dry extract (mg GAE/g d.e.), showed significant variation (Table 2). The domestic Serbian variety under control conditions had the highest TPC (68.16 ± 0.457 mg GAE/g d.e.), significantly greater than all other samples (*p* < 0.05). The Springer mix variety under control conditions followed with 66.84 ± 0.396 mg GAE/g d.e. The lowest TPCwas observed in the Springer mix variety under red nets (53.11 ± 0.229 mg GAE/g d.e.), suggesting that red nets may suppress phenolic accumulation in this variety. The domestic Serbian variety consistently showed higher TPC across all treatments compared to the Springer mix, except under blue nets, where values were similar.

### 2.3. Total Flavonoid Content

The total flavonoid content (TFC) in pot marigoldextracts was determined spectrophotometrically using the aluminum chloride (AlCl_3_) colorimetric method, as described by Lin and Tang [19] with modifications according to Stanojević et al. [20]. TFC, expressed as mg rutin equivalents per gram of dry extract (mg RE/g d.e.), also varied significantly (Table 2). The domestic Serbian variety under red nets exhibited the highest TFC (55.55 ± 0.216 mg RE/g d.e.), significantly higher than other treatments (*p* < 0.05). The domestic variety under control conditions followed with 54.84 ± 0.081 mg RE/g d.e. The lowest TFCwas recorded in the Springer mix variety under blue nets (36.07 ± 0.294 mg RE/g d.e.). The domestic Serbian variety outperformed the Springer mix in flavonoid accumulation across all treatments, particularly under red and control conditions, highlighting the influence of light quality and variety on flavonoid biosynthesis. Light conditions directly influence the production of bioactive compounds, including flavonoids, polyphenols, triterpenoids, and carotenoids, which are responsible for the plant’s antioxidant properties. In field conditions, photoselective shade nets can filter and scatter specific light wavelengths to modify the light spectrum reaching the plants, influencing their physiological responses [14].

### 2.4. Antioxidant Activity

#### 2.4.1. Radical Scavenging Activity-DPPH

The DPPH assay, expressed as EC_50_ values (mg/mL), indicates the concentration required to inhibit 50% of DPPH radicals, with lower values reflecting higher antioxidant activity. The ‘domestic’ Serbian variety under blue nets exhibited the lowest EC_50_ (0.065 ± 0.0021 mg/mL), indicating the highest antioxidant activity (*p* < 0.05). The ‘Springer mix’ variety under blue nets followed with an EC_50_ of 0.071 ± 0.0014 mg/mL. The highest EC_50_, indicating the lowest antioxidant activity, was observed in the ‘Springer mix’ variety under pearl nets (0.102 ± 0.0010 mg/mL). The ‘domestic’ Serbian variety consistently showed lower EC_50_ values across all treatments, particularly under blue nets, suggesting enhanced radical scavenging capacity (Table 3).

Interestingly, the strongest antioxidant activities (blue nets) were not necessarily associated with the highest phenolic or flavonoid content, as observed in previous analyses. This suggests that other bioactive compounds, synergistic effects, or specific phenolic profiles may contribute significantly to antioxidant capacity.

#### 2.4.2. FRAP Assay

The FRAP assay, which quantifies ferric-reducing antioxidant power (mg Fe^2+^/g dry extract), showed highly significant differences among treatments (Table 3). The ‘domestic’ Serbian variety cultivated under pearl nets exhibited the highest FRAP value (53.44 ± 0.197 mg Fe^2+^/g d.e.), significantly surpassing all other treatments (*p* < 0.05). This was closely followed by the same variety grown under control (open-field) conditions, with a value of 51.91 ± 0.260 mg Fe^2+^/g d.e. In contrast, the lowest FRAP activity was recorded in the domestic variety under blue nets (41.75 ± 0.170 mg Fe^2+^/g d.e.). Overall, the ‘domestic’ Serbian variety consistently demonstrated superior antioxidant potential compared to the ‘Springer mix’, except for the blue net treatment, where the two varieties showed similar levels of FRAP activity.

### 2.5. PCA

To explore the relationships among phytochemical and antioxidant variables and the distribution of samples, a PCA was conducted on TEM, TPC, total TFC, DPPH EC_50_, and FRAP values across the eight samples. The first two principal components accounted for 79.79% of the total variance (Factor 1: 50.36%, Factor 2: 29.43%), providing a robust representation of the data structure.

The projection of variables on the factor-plane (Factor 1 × Factor 2) elucidates the underlying relationships among TEM, TPC, TFC, DPPH EC_50_, and FRAP (Figure 1A). Factor 1, which explains 50.36% of the total variance, is strongly defined by negative loadings for flavonoid content (TFC, −0.8527), FRAP (−0.7322), total phenolic content (TPC, −0.6158), and DPPH EC_50_ (−0.6477), along with a pronounced positive loading for TEM (0.6752). This indicates that samples with higher levels of phenolic and flavonoid compounds, along with stronger antioxidant capacity (indicated by lower DPPH EC_50_ values), are located on the negative side of Factor 1, whereas samples with elevated TEM values are positioned on its positive side.

Factor 2, accounting for an additional 29.43% of the variance, is primarily influenced by a strong positive loading for DPPH EC_50_ (0.7013) and a negative loading for TEM (−0.6333), with TPC (−0.5916) and FRAP (−0.3971) also contributing negatively. This suggests that samples exhibiting lower antioxidant activity (higher DPPH EC_50_) are positioned on the positive side of Factor 2, whereas those with higher TEM, TPC, and FRAP values align with its negative axis.

The near-orthogonal positioning of DPPH EC_50_ and TFC on the factor-plane indicates minimal correlation between these two parameters. In contrast, the close clustering of TPC, TFC, and FRAP reflects their mutual association and their collective contribution to the antioxidant potential of the samples

The projection of cases on the factor-plane (Factor 1 × Factor 2) reveals the spatial distribution of the eight pot marigold samples based on their phytochemical composition and antioxidant capacity (Figure 1B). The ‘domestic’ Serbian variety and the ‘Springer mix’ grown under blue nets are positioned on the positive side of Factor 1 and between the negative and slightly positive region of Factor 2. This positioning corresponds to their elevated total extractable matter (TEM: 30.90 and 31.23 g/100 g d.m., respectively) and pronounced antioxidant activity, as indicated by low DPPH EC_50_ values (0.065 and 0.071 mg/mL).

Samples cultivated under control (open-field) conditions—both domestic and ‘Springer mix’ varieties—are grouped on the negative side of both axes. The domestic variety recorded higher TPC (68.16 mg GAE/g d.e.) and TFC (54.84 mg RE/g d.e.) compared to the ‘Springer mix’ variety (TPC: 66.84 mg GAE/g d.e.; TFC: 39.36 mg RE/g d.e.), with both varieties showing moderate and comparable antioxidant activity (DPPH EC50: 0.079 and 0.080 mg/mL, respectively). These results indicate that the domestic Serbian variety was more effective in accumulating bioactive compounds under control conditions.

The domestic variety under pearl nets is located near the origin of Factor 1 and on the positive side of Factor 2, a position influenced by its notably high FRAP value (53.44 mg Fe^2+^/g d.e.) and moderate TFC (50.98 mg RE/g d.e.). Similarly, the ‘Springer mix’ under pearl nets and the ‘domestic’ variety under red nets are positioned on the negative side of Factor 1 and the positive side of Factor 2, associated with high flavonoid content (45.10 and 55.55 mg RE/g d.e.) but comparatively lower TEM (22.29 and 25.40 g/100 g d.m.).

The ‘Springer mix’ under red nets is distinctly separated on the far positive end of Factor 2, reflecting its reduced phytochemical richness, specifically lower TPC (53.11 mg GAE/g d.e.), TFC (37.10 mg RE/g d.e.), and the weakest antioxidant activity among the samples (DPPH EC_50_: 0.101 mg/mL).

Overall, this distribution underscores the influence of blue netting in enhancing extractability and antioxidant efficacy, particularly in the ‘domestic’ Serbian variety, while highlighting the beneficial effect of control (open-field) conditions in promoting phenolic and flavonoid accumulation.

## 3. Discussion

This study provides compelling evidence that colored shade nets substantially modulate the phytochemical profile and antioxidant activity ofpot marigold, with the ‘domestic’ Serbian variety consistently exhibiting superior performance compared to the Italian ‘Springer mix’. Among all light treatments, blue nets were most effective in enhancing TEM across both varieties. The ‘Springer mix’ under blue nets achieved the highest TEM value overall (31.23 ± 0.237 g/100 g d.m.), followed by the domestic variety under blue nets (30.90 ± 0.118 g/100 g d.m.). In comparison, pearl nets yielded the lowest TEM values (22.29 and 24.18 g/100 g d.m., respectively), while red nets and control conditions produced intermediate results. The domestic variety under open-field (control) conditions accumulated the highest TPC (68.16 ± 0.457 mg GAE/g d.e.), while red nets maximized TFC (55.55 ± 0.216 mg RE/g d.e.), Table 2.

In the study by Breda et al. [10], the total phenolic content was significantly lower (~17 mg GA g^−1^ DW for total phenols) than in the pot marigold extract with or without shading in our investigation. In the study by Filipović et al. [21], conducted under Serbian climatic conditions, a higher TPC was recorded in the ‘Plamen Plus’ cultivar (40.9 mg GAE/g), while the ‘Domaći oranž’ cultivar showed a lower value-approximately 24.4% less, averaging 30.8 mg GAE/g over a five-year period under varying levels of compost fertilization.

The variation in phenolic content within the same plant species can be attributed not only to the choice of extraction solvent, but also to several other factors, including environmental conditions (such as light, temperature, and altitude), soil characteristics, and the plant’s stage of maturity. In a study comparing three different extraction solvents, ethanol, ethyl acetate, and n-hexane, ethanol was identified as the most effective for extracting phenols and flavonoids from pot marigold [22].

The phenolic content values obtained in our study are considerably higher than those reported by Veličković et al. [23], who analyzed three different pot marigold extracts (water, ethanol, and a 50:50 hydroethanolic solution), with results of 45.13 mg GAE/g DM, 31.86 mg GAE/g DM, and 29.79 mg GAE/g DM, respectively. In comparison, the study by Breda et al. [10] recorded a significantly lower TPC (~17 mg GAE/g DW) than that found in pot marigold extracts, both with and without shading, in our investigation.

Antioxidant activity followed similar trends: the strongest DPPH radical scavenging was observed in the ‘domestic’ variety under blue nets (EC_50_ 0.065 ± 0.0021 mg/mL), while pearl nets yielded the highest FRAP value (53.44 ± 0.197 mg Fe^2+^/g d.e.), Table 3.

These findings confirm the effectiveness of spectral manipulation via colored nets as a strategy for enhancing bioactive compound production and suggest that the ‘domestic’ Serbian variety holds particular promise for high-value applications in nutraceuticals, pharmaceuticals, and functional foods. These mechanisms were chosen because they are standard methods for assessing a sample’s overall antioxidant capacity by measuring its ability to neutralize free radicals and reduce ferric ions. Radical scavenging activity (like with DPPH) shows a compound’s ability to directly neutralize unstable free radicals, while reducing power (like with the ferric reducing assay, or FRAP) indicates a substance’s ability to donate electrons and reduce Fe^3+^ to Fe^2+^, which are two complementary aspects of a substance’s antioxidant function [24].

Antioxidants, particularly phenolic compounds, play a crucial role in plant stress tolerance by directly scavenging reactive oxygen species (ROS) that cause cellular damage under abiotic (e.g., drought, high light) and biotic (e.g., pathogen, herbivore) stresses.

Phenolics also act as signaling molecules, regulate stress-related gene expression, fortify cell walls, and attract pollinators, thereby enhancing a plant’s ability to adapt to and survive in adverse environmental conditions. Results by Tavasoli et al. [25] represent a significant advance in understanding the level of tolerance and the responsive mechanisms to drought stress in pot marigold during the most critical stage of its life cycle, when reduced rainfall usually occurs in its cultivation area.

The observed responses are largely attributable to the regulatory effects of light quality on secondary metabolism. Blue light, enriched under blue nets, is known to activate phenylpropanoid pathways and antioxidant defenses, which may explain the elevated TEM and enhanced DPPH activity-especially in the domestic variety (Case 3, Figure 1B). Red light, filtered through red nets, appears to promote flavonoid biosynthesis, likely through its influence on specific transcription factors and enzymes involved in flavonoid metabolism. These results partially align with prior findings in other medicinal and aromatic plants, where spectral shifts induced by netting altered the accumulation of key phytochemicals [16,18]. However, unlike those studies, total phenolic content in pot marigold was highest under control (full-spectrum, high-intensity) conditions, suggesting that UV exposure and photoinhibitory stress may be primary drivers of phenolic biosynthesis in this species. Colored nets, while reducing TPC, selectively enhanced other bioactive parameters such as TEM, flavonoids, and radical scavenging activity, demonstrating pathway-specific metabolic regulation.

In the study by Balázs et al. [26], the total carotenoid content in flowers grown under shade was higher than in those exposed to full sunlight (1.154 and 0.872 mg/g in petals, respectively), while no differences were observed in the proportion of individual carotenoids. Among the 29 identified components, in addition to lutein, the main 5,8-epoxycarotenoids were flavoxanthin, chrysanthemaxanthin, and luteoxanthin epimers [27]. These findings suggest that moderate shading may favor the accumulation of total carotenoids without altering their qualitative composition.

Understanding the complex interplay between light signals and metabolic pathways-particularly those governing flavonoid biosynthesis-is critical for the rational design of cultivation systems aimed at maximizing phytochemical output. Future studies focusing on gene expression, enzyme activity, and light receptor signaling could provide deeper mechanistic insights and support targeted biofortification efforts. Importantly, optimizing light conditions not only enhances yield and quality in cultivated medicinal plants but also contributes to the conservation and sustainable use of species with limited natural populations. As supported by Idris et al. [28], light serves not only as an energy source but also as a vital environmental cue that shapes the biosynthetic landscape of secondary metabolites.

Pearl nets, which scatter and homogenize the light spectrum, notably enhanced the ferric-reducing antioxidant power (FRAP) in the ‘domestic’ Serbian variety (Case 5; 53.44 mg Fe^2+^/g d.e.). This suggests that a diffused, spectrally balanced light environment supports the accumulation of reductive compounds. In contrast, the ‘Springer mix’ showed limited responsiveness to this treatment, as well as under red nets (Cases 6 and 8), implying genotypic differences in sensitivity to spectral cues. These results highlight light quality not merely as a passive environmental factor, but as a strategic cultivation tool that can be used to direct specific metabolic outcomes.

Across treatments, the ‘domestic’ Serbian variety consistently outperformed the ‘Springer mix’ in all major bioactive parameters such as TPC, TFC, and antioxidant capacity (DPPH, FRAP). Under open-field conditions (Case 1), it exhibited the highest TPC (68.16 mg GAE/g d.e.), and under red nets (Case 7), the highest TFC (55.55 mg RE/g d.e.), both markedly exceeding those of the ‘Springer mix’ (Case 2: 66.84 mg GAE/g d.e.; 39.36 mg RE/g d.e.). This superiority may reflect genetic adaptations of the domestic variety to regional environmental pressures, particularly Serbia’s moderate, continental climate and fertile soils, which are known to favor secondary metabolite production. Although the ‘Springer mix’ achieved the highest extractive matter under blue nets (Case 4: 31.23 g/100 g d.m.), its overall phenolic and flavonoid content remained consistently lower—likely due to varietal origin, breeding objectives, and ecological mismatch. These findings underscore the domestic variety’s agronomic and phytochemical potential, positioning it as a strong candidate for sustainable, high-value cultivation.

The multivariate PCA analysis reinforced these observations. Factor 1 (50.36% of total variance) distinguished samples with elevated TPC, FC, and FRAP on the negative axis—mainly those under control or red net conditions (e.g., Cases 1 and 2)—from those with high TEM on the positive axis, such as the blue net treatments (Cases 3 and 4). Factor 2 (29.43%) further separated samples based on antioxidant activity, with high DPPH EC_50_ (lower antioxidant potential) positioned on the positive side (e.g., Case 6), and samples with robust phytochemical profiles on the negative side. The tight clustering of TPC, TFC, and FRAP in the variable projection (Figure 1A) suggests coordinated metabolic regulation and their collective contribution to antioxidant functionality, consistent with prior studies [16,17,18].

Notably, the ‘domestic’ Serbian variety occupied quadrants of the factor plane associated with superior biochemical traits, regardless of treatment. This confirms its inherent phytochemical advantage and responsiveness to light modulation. These PCA insights not only validate the treatment effects but also offer a robust analytical framework for optimizing cultivation strategies based on both genetic and environmental inputs.

The application of photoselective shade nets provides the plants with more optimal growth conditions [14]. Plants grown under pearl nets had significantly higher total phenols, flavonoids, and antioxidant properties than non-shading plants [16]. It seems that the response of plants is different to various light spectra [14]. These results show that plant response to the increase in antioxidant capacity varies with the type of photoselective net. The increase in antioxidant levels after the manipulation of environmental light using shade nets can improve the quality of pot marigold. Light intensity affects the antioxidant activity of marigold, with both UV-B radiation and shading influencing its phytochemical composition. Specifically, UV-B radiation and shading can lead to an increase in certain antioxidant compounds, but the effects are complex and depend on the specific light wavelengths and treatments applied. Studies show that red and blue shade nets can alter the light balance, affecting antioxidant capacity and pigments differently. Changes in light intensity can influence the production of phenolic compounds and other antioxidants in marigold flowers. Red shade nets can enhance the plant’s ability to absorb photosynthetically active light. The antioxidant content in pot marigold extract was found to be 5.59 g GAE/100 g of dry weight in a study bystudy Savić-Gajić et al. [27] Its antioxidant activity was expressed based on a half-maximal inhibitory concentration which was found to be EC −0.096 mg/mL. The values reported in the literature are similar to ours or slightly lower.

This study provides compelling evidence that light quality, modulated through colored shade nets, significantly influences the phytochemical profile and antioxidant capacity of pot marigold, with particularly strong effects observed in the ‘domestic’ Serbian variety. High levels of TPC), TFC, and antioxidant activity under control, red, and blue net treatments highlight the synergy between Serbia’s agro-ecological conditions and targeted light manipulation. These findings suggest that Serbia offers a favorable environment for cultivating high-quality pot marigold suitable for pharmaceutical, nutraceutical, cosmetic, and functional food applications.

The superior performance of the ‘domestic’ Serbian variety, especially in TPC, TFC, and antioxidant parameters, positions it as a promising candidate for high-value product development. Its responsiveness to light modulation underscores its potential in controlled-environment agriculture, where tailored spectral conditions can be used to direct secondary metabolite biosynthesis.

Future research should delve deeper into the molecular and physiological mechanisms by which different light spectra influence metabolite accumulation. Investigating the role of additional net colors (e.g., green, black), seasonal dynamics, and genotype-environment interactions could further optimize phytochemical yield. Moreover, chemical profiling of individual phenolic and flavonoid constituents responsible for antioxidant activity will expand the industrial relevance of the crop.

The current findings lay a strong foundation for the strategic cultivation of pot marigold in Serbia. With increasing global demand for plant-based antioxidants and natural bioactives, Serbia is well-positioned to emerge as a regional hub for the sustainable production of high-quality calendula raw materials for both domestic and international markets.

This study provides evidence for improving and developing optimal conditions to increase yield, enhance active compounds, and improve the efficacy quality of the pot marigold. In addition, this research highlights practical implications for agricultural practices and potential nutraceutical/pharmaceutical applications.

## 4. Materials and Methods

### 4.1. Plant Material and Growing Conditions

The experiment with cultivated pot marigold (*Calendula officinalis* L.) (from ‘domestic’ variety cultivated in Serbia and ‘Springer mix’, obtained from Italy,) was carried out in an experimental garden in the village of Moravac in south Serbia (21°42’ E, 43°30’ N, altitude 159 m a.s.l.) between 2023 and 2024.

The results of the agrochemical analysis of the alluvial soils type from Table 4. show that soil is highly accumulative with a fairly high humus content of 3.59% and very well supplied with phosphorus and potassium (40 mg/100 g), Table 4.

Seeds of two marigold cultivars, Mix (Springer) and Domaćiorange, were sown in early April 2023 in containers with 74 cells (4 cm diameter × 4 cm depth). Seedlings were raised in Pindstrup substrate formulated for vegetable production. After sowing and watering, containers were placed in a plastic tunnel for germination. Emergence occurred 7–10 days after sowing, with 89% germination in Mix (Springer) and 84% in Domaći orange. During the nursery stage, irrigation and foliar fertilization (prior to transplanting) were applied; no plant protection was required due to good plant health.

Standard agrotechnical practices included autumn plowing and application of mineral fertilizer (NPK 12:12:17 + 2% Mg + 14% S + 0.02% B + 0.01% Zn, Elixir Zorka Supreme) prior to seedbed preparation, incorporated by rototilling. The experiment was arranged in a randomized block design with three replications, each plot consisting of 30 plants. Treatments included shading with photoselective nets (40% shade index; pearl, red, and blue), while unshaded plants served as the control. Transplanting was carried out around May 20 at the 4–5 leaf stage, with 45 × 15 cm spacing in a zigzag pattern to optimize land use.

During the growing season, inter-row cultivation was performed for weed control and soil loosening, and plants were irrigated by drip system. Powdery mildew (Podosphaeraxanthii (Castagne)) was recorded. As no fungicides are registered for marigold powdery mildew control in Serbia, a microbial fertilizer (Microbacilus mix) was applied foliarly several times during the season. Thrips were also observed as pests. Flower harvesting started in early July.

Seeds of two marigold cultivars, Mix (Springer) and Domaći orange, were sown in 74-cell containers (4 × 4 cm) in early April 2023, using Pindstrup substrate intended for vegetable crops. Germination occurred 7–10 days after sowing, with rates of 89% and 84% for Mix and Domaći orange, respectively. During the nursery stage, seedlings were watered and foliar fertilized prior to transplanting, without the need for plant protection measures.

### 4.2. Sample Preparation

Dried aerial parts of pot marigold (5 g) from each treatment and variety were extracted using 100 mL of 50% ethanol (*v*/*v*) at a solvent-to-sample ratio of 1:20 (m/V). The extraction was performed at room temperature with constant agitation for 24 h. All extraction procedures were independently performed in triplicate (*n* = 3). The resulting extracts were filtered, and 3 mL of each extract was evaporated to dryness to determine the total extractive matter (TEM) yield, expressed as g/100 g of dry plant material. The remaining extracts were stored at 4 °C until further analysis

### 4.3. Determination of Total Extractive Matter (TEM)

The TEM was calculated by evaporating 3 mL of each extract to dryness in pre-weighed containers. The dry residue was weighed, and the yield was calculated as grams of extract per 100 g of dry plant material (g/100 g p.m.). Measurements were performed in triplicate, and results were expressed as mean ± standard deviation.

### 4.4. Total Phenolic Content

The total phenolic content (TPC) in pot marigoldextracts was determined using the Folin-Ciocalteu method, as described by Singleton and Rossi [29] and modified by Stanojević et al. [20]. A 0.5 mL aliquot of each extract (0.5 mg/mL) was mixed with 2.5 mL of Folin-Ciocalteu reagent (diluted 1:10 with distilled water) and 2 mL of 7.5% sodium carbonate solution. The mixture was incubated at room temperature for 30 min, and absorbance was measured at 765 nm using a UV-Vis spectrophotometer. Phenolic content was calculated using a gallic acid standard curve (A_765_ = 0.01528 + 5.04832*c, where c is the concentration in mg/mL) and expressed as mg gallic acid equivalents per gram of dry extract (mg GAE/g d.e.) and per gram of dry plant material (mg GAE/g p.m.). Measurements were performed in triplicate.

### 4.5. Total Flavonoid Content

Total flavonoid content (TFC) was determined using the aluminum chloride method. A 0.5 mL aliquot of each extract (0.5 mg/mL) was mixed with 1.5 mL of 95% ethanol, 0.1 mL of 10% aluminum chloride, 0.1 mL of 1 M potassium acetate, and 2.8 mL of distilled water. After 30 min of incubation at room temperature, absorbance was measured at 415 nm. Flavonoid content was calculated using a rutin standard curve (A_415_ = 0.0461 + 14.171*c, where c is the concentration in mg/mL) and expressed as mg rutin equivalents per gram of dry extract (mg RE/g d.e.) and per gram of dry plant material (mg RE/g p.m.), Stanojević et al. [20]. Measurements were performed in triplicate.

### 4.6. DPPH Antioxidant Activity

The antioxidant activity was evaluated using the 2,2-diphenyl-1-picrylhydrazyl (DPPH) radical scavenging assay. Extracts were diluted to concentrations of 0.5, 0.25, 0.125, 0.0625, 0.03125, and 0.015625 mg/mL. The ethanolic solution of DPPH radical (1 cm^3^, 3 × 10^−4^ mol/dm^3^) was added to 2.5 cm^3^ of extract solutions of different concentrations. The reaction mixtures were incubated at room temperature, in the dark, for 20 min, after which the absorbance was measured at 517 nm (A_S_). The absorbance at 517 nm was determined both for a pure ethanolic solution of DPPH radical diluted in the specified ratio (1 cm^3^ of DPPH radicals of a given concentration + 2.5 cm^3^ of ethanol) (A_C_), and for extracts before treatment with DPPH radical (2.5 cm^3^ of extract + 1 cm^3^ of ethanol) (A_B_). Ethanol was used as a blank.

The degree of neutralization of free radicals is calculated according to the following formula [20]:$$\mathrm{Degree}\;\mathrm{of}\;\mathrm{DPPH}\;\mathrm{radical}\;\mathrm{neutralisation}\;(\%)\;=\;100-\left[\left(A_{S}-A_{B}\right)\times \frac{100}{A_{C}}\right]$$

*A_S_* = Absorbance of the “sample” at 517 nm. “Sample”—ethanolic solution of the extract treated with a solution of DPPH radicals.

*A_B_* = Absorbance of the “blank” at 517 nm. “Blank”—ethanol solution of the extract that has not been treated with a solution of DPPH radicals.

*A_C_* = Absorbance of the “control” at 517 nm. “Control”—ethanolic solution of DPPH radicals (diluted in a ratio of 1 cm^3^ of DPPH radicals with a concentration of 3 × 10^−4^ mol/dm^3^ + 2.5 cm^3^ of ethanol).

The EC_50_ value (mg/mL), representing the concentration required to inhibit 50% of DPPH radicals, was determined using linear regression (y = a + b*x). Measurements were performed in triplicate.

### 4.7. FRAP Assay

The ferric-reducing antioxidant power (FRAP) as described by Benzie and Strain [30] was used to assess the reducing capacity of the extractsA 0.5 mL aliquot of each extract (0.5 mg/mL) was mixed with 2.5 mL of phosphate buffer (pH 6.6) and 2.5 mL of 1% potassium ferricyanide. The mixture was incubated at 50 °C for 20 min, followed by the addition of 2.5 mL of 10% trichloroacetic acid. After centrifugation, 2.5 mL of the supernatant was mixed with 2.5 mL of distilled water and 0.5 mL of 0.1% ferric chloride. Absorbance was measured at 593 nm. The FRAP value was calculated using a ferrous sulfate standard curve (A_593_ = 0.00495 + 0.65743*c, where c is the concentration in mg/mL) and expressed as mg Fe^2+^ per gram of dry extract (mg Fe^2+^/g d.e.) and mmol Fe^2+^ per gram of dry extract (mmol Fe^2+^/g d.e.). Measurements were performed in triplicate.

### 4.8. Statistical Analysis

All measurements were conducted in triplicate, and results were expressed as mean ± standard deviation. Data were analyzed using two-way analysis of variance (ANOVA) followed by Duncan’s multiple range test to determine significant differences among samples (*p* < 0.05). The LSD test was applied to determine the significance of parameters related to yield and yield components of pot marigold. Statistical analyses including PCA were performed using STATISTICA 14 TIBCO Software, Inc., Palo Alto, CA, USA) (2020). Data Science Workbench, version 14.).

## 5. Conclusions

Pot marigold is a well-established medicinal plant known for its pronounced antioxidant and anti-inflammatory properties, primarily driven by its rich content of phenolic and flavonoids compounds. In this study, the ‘domestic’ Serbian variety demonstrated a particularly robust phytochemical profile, with TPC reaching 68.21 mg GAE/g d.e. and TFC of 55.55 mg RE/g d.e. Such elevated levels highlight its potential not only for pharmacological and nutraceutical applications but also for positioning Serbia as a strategic producer of high-quality pot marigold raw material for the global market. The antioxidant activity, evaluated through DPPH and FRAP assays, confirmed the strong bioactivity of this variety. Notably, the lowest EC_50_ value (0.065 mg/mL) was observed under blue net cultivation, indicating enhanced radical-scavenging efficiency under modified light conditions. These findings reinforce the central role of phenolic and flavonoid compounds in mediating antioxidant effects and emphasize the importance of environmental modulation-particularly light quality-as a key factor in optimizing secondary metabolite production.

## Figures and Tables

**Figure 1 plants-14-03512-f001:**
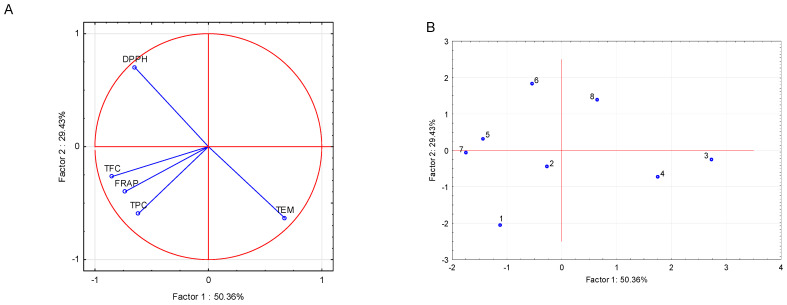
Projection of the variables (**A**) and Cases (**B**) on the factor-plane (Factor 1 × Factor 2) for *Calendula officinalis* samples. 1-‘Domestic’, Serbia, Control; 2-‘Springer mix’, Italy, Control; 3-‘Domestic’, Serbia, Blue; 4-‘Springer mix’, Italy, Blue; 5-‘Domestic’, Serbia, Pearl; 6-‘Springer mix’, Italy, Pearl; 7-‘Domestic’, Serbia, Red; 8-‘Springer mix’, Italy, Red.

**Table 1 plants-14-03512-t001:** Morphological characteristics, yield parameters and total yield.

Variety	Treatment	Plant Height (cm)	Mass of the Whole Plant (g)	Number of Flowers per Plant	The Average Mass of the Flower (g)	Flower Diameter (cm)	Flower Fresh Yield (t/ha)	Flower Dry Yield (t/ha)
’Springer mix’	Pearl	57.5	219.6	54.18	1.006	5.012	13.537	2.622
	Red	58.2	250.3	59.77	1.036	5.437	15.310	2.990
	Blue	57.1	216.2	44.09	0.932	4.983	10.610	2.048
	Control	45.9	178.9	51.40	0.936	4.509	12.117	2.353
LSD 0.05		3.583	26.86	5.490	0.2825	0.413	2.326	0.415
LSD 0.01		5.429	40.70	8.319	0.4281	0.627	3.524	0.683
‘domestic, Serbia’	Pearl	48.3	188.2	40.88	0.88	4.405	9.032	1.714
	Red	51.2	190.6	46.94	0.85	4.550	10.047	1.902
	Blue	48.6	189.4	39.01	0.77	4.308	7.507	1.467
	Control	38.0	153.4	42.57	0.7	4.177	8.102	1.516
LSD 0.05		3.666	17.84	5.193	0.0893	0.236	1.304	0.2527
LSD 0.01		5.555	27.03	7.877	0.1354	0.358	1.976	0.3829
A*XB** LSD 0.05		3.369	18.510	5.020	0.0783	0.332	1.577	0.3083
LSD 0.01		4.676	25.701	6.968	0.1087	0.461	2.189	0.4279

* A-cultivar, ** B-photoselective nets.

**Table 2 plants-14-03512-t002:** Total extractive matter (TEM), Total phenol content (TPC) and Total flavonoid content (TFC) of pot marigoldvarieties under different color nets.

Variety	Shade Net	TEM (g/100 g d.m.)	TPC mg GAE/g d.e.	TFC mg RE/g d.e.
‘domestic’, Serbia	control	30.0 ± 0.227 ^c^	68.2 ± 0.457 ^a^	54.8 ± 0.081 ^b^
‘Springer mix’	control	25.2 ± 0.219 ^e^	66.8 ± 0.396 ^b^	39.4 ± 0.141 ^e^
‘domestic’, Serbia	blue	30.9 ± 0.118 ^b^	53.8 ± 0.396 ^g^	37.5 ± 0.163 ^f^
‘Springer mix’	blue	31.2 ± 0.237 ^a^	55.0 ± 0.396 ^f^	36.1 ± 0.294 ^h^
‘domestic’, Serbia	pearl	24.2 ± 0.399 ^f^	56.8 ± 0.825 ^e^	51.0 ± 0.216 ^c^
‘Springer mix’	pearl	22.3 ± 0.381 ^g^	57.6 ± 0.605 ^d^	45.1 ± 0.163 ^d^
‘domestic’, Serbia	red	25.2 ± 0.151 ^e^	63.1 ± 0.457 ^c^	55.5 ± 0.216 ^a^
‘Springer mix’	red	27.5 ± 0.365 ^d^	53.1 ± 0.229 ^h^	37.1 ± 0.141 ^g^
	variety	**	**	**
	netting	**	**	**
	interaction	**	**	**

Different superscript letters (a–h) indicate significant differences within each column (*p* < 0.05, Duncan’s test). Statistical significance for effects: ** *p* < 0.01.

**Table 3 plants-14-03512-t003:** Antioxidant activity (DPPH and FRAP) of pot marigold varieties under different color nets.

Variety	Color Nets	DPPH EC_50_, mg/mL	FRAP (mg Fe^2+^/g d.e.).
‘domestic’, Serbia	control	0.0794 ± 0.0003 ^c^	51.9 ± 0.260 ^f^
‘Springer mix’	control	0.0801 ± 0.0005 ^c^	48.7 ± 0.170 ^d^
‘domestic’, Serbia	blue	0.0651 ± 0.0021 ^a^	41.7 ± 0.170 ^a^
‘Springer mix’	blue	0.0705 ± 0.0014 ^b^	48.8 ± 0.429 ^d^
‘domestic’, Serbia	pearl	0.0928 ± 0.0004 ^e^	53.4 ± 0.197 ^g^
‘Springer mix’	pearl	0.1015 ± 0.0010 ^f^	44.7 ± 0.260 ^b^
‘domestic’, Serbia	red	0.0961 ± 0.0015 ^d^	50.4 ± 0.260 ^e^
‘Springer mix’	red	0.1014 ± 0.0010 ^f^	47.2 ± 0.295 ^c^
	variety	**	**
	netting	**	**
	intteraction	**	**

Different superscript letters (a–g) indicate significant differences within each column (*p* < 0.05, Duncan’s test). Statistical significance for effects: ** *p* < 0.01.

**Table 4 plants-14-03512-t004:** Chemical analysis of soil (0–30 cm dept).

pH in 1 M KCl	pH in H_2_O	CaCO_3_ %	Humus %	N %	P_2_O_5_	K_2_O
					mg/100 g	
6.62	7.54	0.69	3.59	0.18	40	40

## Data Availability

The authors confirm that the datasets generated during and/or analyzed during the current study are available from the corresponding author on reasonable request.

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
