# Peer review of "Phytochemical Composition and Antioxidant Activity of Pot Marigold (Calendula officinalis): The Impact of Light Modification"

_plants, 2025, doi:10.3390/plants14223512_

Round 1

Reviewer 1 Report

Comments and Suggestions for Authors

Phytochemical composition and antioxidant activity of pot marigold (Calendula officinalis): The impact of light modification

This study explored the impact of light modification on growing the pot marigold. The different color light during growing and the plant variety affected the yield, phytochemical composition, and antioxidant activity. The experiment design is well conducted with clear objectives relating to the impact of light-modified environments. This study provides evidence for improving and developing optimal conditions to increase yield, enhance active compounds, and improve the efficacy quality of the pot marigold. In addition, this research highlights practical implications for agricultural practices and potential nutraceutical/pharmaceutical applications. With minor revisions and clarification on certain points, it is suitable for publication in its target journal.

  • Lines 295-298: It would be better to discuss more in the discussion session about the probable bioactive compounds that exhibit the antioxidant activity. Are there any previous reports?
  • Lines 402-404: The paragraph is a repeat of lines 309-401

Author Response

This study explored the impact of light modification on growing the pot marigold. The different color light during growing and the plant variety affected the yield, phytochemical composition, and antioxidant activity. The experiment design is well conducted with clear objectives relating to the impact of light-modified environments. This study provides evidence for improving and developing optimal conditions to increase yield, enhance active compounds, and improve the efficacy quality of the pot marigold. In addition, this research highlights practical implications for agricultural practices and potential nutraceutical/pharmaceutical applications. With minor revisions and clarification on certain points, it is suitable for publication in its target journal.

  • Lines 295-298: It would be better to discuss more in the discussion session about the probable bioactive compounds that exhibit the antioxidant activity. Are there any previous reports?

Light conditions directly influence the production of bioactive compounds, including flavonoids, polyphenols, triterpenoids, and carotenoids, which are responsible for the plant's antioxidant properties. In field conditions, photoselective shade nets can filter and scatter specific light wavelengths to modify the light spectrum reaching the plants, influencing their physiological responses [24]. 

  • Lines 402-404: The paragraph is a repeat of lines 309-401

Yes..we are exclude repeat  part

This study explored the impact of light modification on growing the pot marigold. The different color light during growing and the plant variety affected the yield, phytochemical composition, and antioxidant activity. The experiment design is well conducted with clear objectives relating to the impact of light-modified environments. This study provides evidence for improving and developing optimal conditions to increase yield, enhance active compounds, and improve the efficacy quality of the pot marigold. In addition, this research highlights practical implications for agricultural practices and potential nutraceutical/pharmaceutical applications. With minor revisions and clarification on certain points, it is suitable for publication in its target journal.

  • Lines 295-298: It would be better to discuss more in the discussion session about the probable bioactive compounds that exhibit the antioxidant activity. Are there any previous reports?

Light conditions directly influence the production of bioactive compounds, including flavonoids, polyphenols, triterpenoids, and carotenoids, which are responsible for the plant's antioxidant properties. In field conditions, photoselective shade nets can filter and scatter specific light wavelengths to modify the light spectrum reaching the plants, influencing their physiological responses [24]. 

  • Lines 402-404: The paragraph is a repeat of lines 309-401

Yes..we are exclude repeat  part

This study explored the impact of light modification on growing the pot marigold. The different color light during growing and the plant variety affected the yield, phytochemical composition, and antioxidant activity. The experiment design is well conducted with clear objectives relating to the impact of light-modified environments. This study provides evidence for improving and developing optimal conditions to increase yield, enhance active compounds, and improve the efficacy quality of the pot marigold. In addition, this research highlights practical implications for agricultural practices and potential nutraceutical/pharmaceutical applications. With minor revisions and clarification on certain points, it is suitable for publication in its target journal.

  • Lines 295-298: It would be better to discuss more in the discussion session about the probable bioactive compounds that exhibit the antioxidant activity. Are there any previous reports?

Light conditions directly influence the production of bioactive compounds, including flavonoids, polyphenols, triterpenoids, and carotenoids, which are responsible for the plant's antioxidant properties. In field conditions, photoselective shade nets can filter and scatter specific light wavelengths to modify the light spectrum reaching the plants, influencing their physiological responses [24]. 

  • Lines 402-404: The paragraph is a repeat of lines 309-401

Yes..we are exclude repeat  part

This study explored the impact of light modification on growing the pot marigold. The different color light during growing and the plant variety affected the yield, phytochemical composition, and antioxidant activity. The experiment design is well conducted with clear objectives relating to the impact of light-modified environments. This study provides evidence for improving and developing optimal conditions to increase yield, enhance active compounds, and improve the efficacy quality of the pot marigold. In addition, this research highlights practical implications for agricultural practices and potential nutraceutical/pharmaceutical applications. With minor revisions and clarification on certain points, it is suitable for publication in its target journal.

  • Lines 295-298: It would be better to discuss more in the discussion session about the probable bioactive compounds that exhibit the antioxidant activity. Are there any previous reports?

Light conditions directly influence the production of bioactive compounds, including flavonoids, polyphenols, triterpenoids, and carotenoids, which are responsible for the plant's antioxidant properties. In field conditions, photoselective shade nets can filter and scatter specific light wavelengths to modify the light spectrum reaching the plants, influencing their physiological responses [24]. 

  • Lines 402-404: The paragraph is a repeat of lines 309-401

Yes..we are exclude repeat  part

This study explored the impact of light modification on growing the pot marigold. The different color light during growing and the plant variety affected the yield, phytochemical composition, and antioxidant activity. The experiment design is well conducted with clear objectives relating to the impact of light-modified environments. This study provides evidence for improving and developing optimal conditions to increase yield, enhance active compounds, and improve the efficacy quality of the pot marigold. In addition, this research highlights practical implications for agricultural practices and potential nutraceutical/pharmaceutical applications. With minor revisions and clarification on certain points, it is suitable for publication in its target journal.

  • Lines 295-298: It would be better to discuss more in the discussion session about the probable bioactive compounds that exhibit the antioxidant activity. Are there any previous reports?

Light conditions directly influence the production of bioactive compounds, including flavonoids, polyphenols, triterpenoids, and carotenoids, which are responsible for the plant's antioxidant properties. In field conditions, photoselective shade nets can filter and scatter specific light wavelengths to modify the light spectrum reaching the plants, influencing their physiological responses [24]. 

  • Lines 402-404: The paragraph is a repeat of lines 309-401

Yes..we are exclude repeat  part

This study explored the impact of light modification on growing the pot marigold. The different color light during growing and the plant variety affected the yield, phytochemical composition, and antioxidant activity. The experiment design is well conducted with clear objectives relating to the impact of light-modified environments. This study provides evidence for improving and developing optimal conditions to increase yield, enhance active compounds, and improve the efficacy quality of the pot marigold. In addition, this research highlights practical implications for agricultural practices and potential nutraceutical/pharmaceutical applications. With minor revisions and clarification on certain points, it is suitable for publication in its target journal.

  • Lines 295-298: It would be better to discuss more in the discussion session about the probable bioactive compounds that exhibit the antioxidant activity. Are there any previous reports?

Light conditions directly influence the production of bioactive compounds, including flavonoids, polyphenols, triterpenoids, and carotenoids, which are responsible for the plant's antioxidant properties. In field conditions, photoselective shade nets can filter and scatter specific light wavelengths to modify the light spectrum reaching the plants, influencing their physiological responses [24]. 

  • Lines 402-404: The paragraph is a repeat of lines 309-401

Yes..we are exclude repeat  part

Reviewer 2 Report

Comments and Suggestions for Authors

The manuscript “Phytochemical composition and antioxidant activity of pot marigold (Calendula officinalis): The impact of light modification” by Lidija and coworkers presents a comparative analysis between two varieties of C. officinalis regarding their phytochemical content, antioxidant performance, and total extractive matter. The retrieved data was utilized to perform a principal component analysis. The strength of the submitted version is the consideration of the influence of light exposure in the mentioned species; however, numerous studies have been conducted for the same purpose considering similar variables. Because of this, this manuscript is not recommended for publication in Plants, but the following comments are advised to improve its quality.

Paragraph 50-53 is short. The authors are recommended to merge it with paragraph from L. 54-58.

The use of abbreviation is inconsistent in L. 67 and 68, since one is defined while the other one is not.

The aim of this study should be expanded by mentioning the main performed experiments, as well as a general background of the retrieved results.

In L. 88, it is unnecessary to mention the complete name of the studied species. The authors are recommended to revise this throughout the manuscript.

  1. 139 is missing the model of the utilized rotavapor.

The results section must be improved by adding figures that represent the obtained results. In the current version, must part of the retrieved information is listed in Tables.

The discussion of the submitted version utilizes several abbreviations that have been already defined in previous section. Therefore, the authors are recommended to revise this to improve the consistency of terms. On the other hand, it utilizes the complete definitions of terms that should be utilized with abbreviations.

In the discussion section, the authors are missing to compare the retrieved numbers from their experiments with other studies where similar experiments have been performed. In the case of the antioxidant performance, the authors must mention what is the importance of these findings and according to current literature, they are advise to assess if the antioxidant activity is high, moderate, or low.

Author Response

Paragraph 50-53 is short. The authors are recommended to merge it with paragraph from L. 54-58.

We create new sentence …

Based on ethnobotanical knowledge from eastern Serbia, the pot marigold flower is most commonly used in etno medicine [7, 8], while in other regions of Serbia, the flowers are traditionally used as a spice in soups, broths, and various dishes, but the leaves as well incorporated into salads [9].

The use of abbreviation is inconsistent in L. 67 and 68, since one is defined while the other one is not.

We replace p with p

The aim of this study should be expanded by mentioning the main performed experiments, as well as a general background of the retrieved results.

These studies will contribute to the development of new techniques aimed at creating optimal microclimatic conditions to increase yield, enhance the content of active compounds, and improve the efficacy quality of marigold.

In L. 88, it is unnecessary to mention the complete name of the studied species. The authors are recommended to revise this throughout the manuscript.

 We adopt …..pot marigold……and replace throughout the manuscript

 is missing the model of the utilized rotavapor.

KA Werke GmbH & Co. KG, Germany

The results section must be improved by adding figures that represent the obtained results. In the current version, must part of the retrieved information is listed in Tables.

We decided to save tables….for better presentation…please for your suggestion!!!

The discussion of the submitted version utilizes several abbreviations that have been already defined in previous section. Therefore, the authors are recommended to revise this to improve the consistency of terms. On the other hand, it utilizes the complete definitions of terms that should be utilized with abbreviations.

Yes, we use pot marigold (and remove all abrrrevations calendula officinalis; C. officinalis; Calendula sp.) during the all manuscript body.

In the discussion section, the authors are missing to compare the retrieved numbers from their experiments with other studies where similar experiments have been performed. In the case of the antioxidant performance, the authors must mention what is the importance of these findings and according to current literature, they are advise to assess if the antioxidant activity is high, moderate, or low.

Light intensity affects the antioxidant activity of marigold, with both UV-B radiation and shading influencing its phytochemical composition. Specifically, UV-B radiation and shading can lead to an increase in certain antioxidant compounds, but the effects are complex and depend on the specific light wavelengths and treatments applied. Studies show that red and blue shade nets can alter the light balance, affecting antioxidant capacity and pigments differently. Changes in light intensity can influence the production of phenolic compounds and other antioxidants in marigold flowers. Red shade nets can enhance the plant’s ability to absorb photosynthetically active ligh.

Antioxidants content in pot marigold extract in study Savić- Gajić et al. [28] of 5.59 g GAE/100 g of dry weight. Its antioxidant activity was expressed based on a half-maximal inhibitory concentration which was found to be EC - 0.096 mg/ml. The values reported in the literature are similar to ours or slightly lower.

  1. Savic Gajic, I.; Savic, I.; Skrba, M.; Dosić, A.; Vujadinovic, D. Food additive based on the encapsulated pot marigold (Calendula officinalisL.) flowers extract in calcium alginate microparticles.  J. Food Process Preserv. 2022, 46(10), e15792.

Reviewer 3 Report

Comments and Suggestions for Authors

The manuscript entitled "Phytochemical composition and antioxidant activity of pot marigold (Calendula officinalis): The impact of light modification" was presented and reviewed. In this review, some corrections and clarifications are proposed to the authors.

Introduction

Line 61: p-coumaric, the p should be in italic.

Line 67: In here the authors mention the relevance of edible flowers as a source of nutritional compounds, mentioning this report. To clarify, are these quantifications mentioned from the edible flower of the Calendula or another part of the plant?

In this section the idea of the use of colored photo-selective shading nets is presented, and it is indicated that the use of colored nets has shown positive influence in other aromatic plants, such as sage. In this example with sage, only pearl nets were used, so the effect of color was not evaluated. Thus, I think that in this section a better clarification should be made regarding the colored nets, namely, what is the direct influence of the shading net color on the plant, and why were these colors considered?

Materials and methods

Line 128-133: Please carefully check this information, since it is repeated here.

Line 135: It should be previously specified how the dried plant material was obtained. This information is not presented, nor is how the plant material was collected.

Line 228: correct SI units should be used.

Table 2: It is mentioned in the table TFC, but this does not correspond with the FC from the caption. Please correct this. Also, the uppercase letters, which indicate statistical significance, should start with A as the highest obtained value.

Please revise the structure of the results sections. The authors have 3.2 for phytochemical content, with only one sub-section. Why didn't you put the total flavonoid content under the 3.2 subject, for example?

Was the plant growth and yield affected by the treatments? Was this evaluated?

Line 389: Please clarify this statement; it is not clear which was identified as the most effective.

Line 414: Please clarify this statement.

Line 474 states that other studies have shown net shading consistently increases the phytochemical content of plants; however, the results presented in this manuscript do not support this claim for TPC. This should be further discussed.

The results show the highest TPC for the control samples, meaning net shading decreases TPC. This seems to contradict the manuscript's argument, so further discussion or restructuring may be needed.

If there is a considerable lack of information regarding the chemical characterization and profiling of this plant species, perhaps this could be the more direct focus of this manuscript, instead of the net shading.

Furthermore, as a comment/question for the authors: If the focus was to compare the shading effect of colored nets against the unshaded open-field cultivation (as mentioned in lines 93-94), why didn't you consider a white shading net as a control for the colored nets (for example)? Because if the unshaded was considered as the control, these plants were exposed to much different conditions (since no shading was applied, regardless of the color). Thus, a white shading net could be used as a control for the shading color.

Author Response

Introduction

Line 61: p-coumaric, the p should be in italic.

Yes we changes p in italic

Line 67: In here the authors mention the relevance of edible flowers as a source of nutritional compounds, mentioning this report. To clarify, are these quantifications mentioned from the edible flower of the Calendula or another part of the plant?

The flowers are edible, while other parts of the plant, such as young leaves, can also be used as food.

In this section the idea of the use of colored photo-selective shading nets is presented, and it is indicated that the use of colored nets has shown positive influence in other aromatic plants, such as sage. In this example with sage, only pearl nets were used, so the effect of color was not evaluated. Thus, I think that in this section a better clarification should be made regarding the colored nets, namely, what is the direct influence of the shading net color on the plant, and why were these colors considered?

Different plant species respond differently to light modified by shading with colored nets. Pearl nets are photo-neutral and are mainly compared with other colored nets. However, this pattern of response cannot be generalized, since the effect of sun radiation is dependent on the plant species. Buthelezi et al., (2016) present in their research that aromatic herbs (coriander, marjoram and oregano) grown under black nets achieve higher antioxidant content than herbs from open field and other shade nets. For example, basil plants grown under blue shade nets from the second harvest are characterized by the highest eugenol content (20.9%) and highest antioxidant activity (efficient concentration - EC50, 0.003 mg mL−1), (Milenković et al., 2019).

Buthelezi, M. N. D., Soundy, P., Jifon, J., & Sivakumar, D. (2016). Spectral quality of photoselective nets improves phytochemicals and aroma volatiles in coriander leaves (Coriandrum sativum L.) after postharvest storage. Journal of Photochemistry and Photobiology B: Biology, 161, 328–334. https://doi.org/10.1016/j. jphotobiol.2016.05.032

Milenković, L., Stanojević, J., Cvetković, D., Stanojević, L., Lalević, D., Šunić, L., Fallik, E., Ilić S.Z.  (2019). New technology in basil production with high essential oil yield and quality. Industrial Crops and Products.140:111718        DOI:10.1016/j.indcrop.2019.111718                                                                                                                                                                                        

Materials and methods

Line 128-133: Please carefully check this information, since it is repeated here.

Yes, you are right, we exclude these sentence.

Line 135: It should be previously specified how the dried plant material was obtained. This information is not presented, nor is how the plant material was collected.

The harvest was carried out manually. Whole flower heads were picked, measured fresh, placed in crates, and dried in a thin layer in a well-ventilated attic space in the shade (almost in darkness). Drying lasted about 8–10 days, depending on the air humidity. Harvesting was done when the inflorescences were fully open and the ray florets were fresh and brightly colored, during sunny and fair weather.

Line 228: correct SI units should be used.

Table 2: It is mentioned in the table TFC, but this does not correspond with the FC from the caption. Please correct this. Also, the uppercase letters, which indicate statistical significance, should start with A as the highest obtained value.

Yes we adopt your suggestion and incorporate in table

Table 3 Total extractive matter (TEM), Total phenol content (TPC) and Total flavonoid content (TFC) of Calendula officinalis varieties under different color nets

Variety

Shade Net

TEM

 (g/100g d.m.)

TPC

mg GAE/g d.e.

TFC

mg RE/g d.e.

ʹdomesticʹ, Serbia

control

30.0± 0.227c

68.2± 0.457a

54.8± 0.081b

ʹSpringer mixʹ

control

25.2± 0.219c

66.8± 0.396b

39.4± 0.141e

ʹdomesticʹ, Serbia

blue

30.9± 0.118b

53.8± 0.396g

37.5± 0.163f

ʹSpringer mixʹ

blue

31.2± 0.237a

55.0± 0.396f

36.1± 0.294h

ʹdomesticʹ, Serbia

pearl

24.2± 0.399f

56.8± 0.825e

51.0± 0.216c

ʹSpringer mixʹ

pearl

22.3± 0.381g

57.6± 0.605d

45.1± 0.163d

ʹdomesticʹ, Serbia

red

25.2± 0.151e

63.1± 0.457c

55.5± 0.216a

ʹSpringer mixʹ

red

27.5± 0.365d

53.1± 0.229h

37.1± 0.141g

variety

**

**

**

netting

**

**

**

interaction

**

**

**

Please revise the structure of the results sections. The authors have 3.2 for phytochemical content, with only one sub-section. Why didn't you put the total flavonoid content under the 3.2 subject, for example?

We revised title and sub-title……

Was the plant growth and yield affected by the treatments? Was this evaluated?

Overall, it was found that the application of photoselective shade nets provide the plants with more optimal growth conditions than when they were not used. It seems that the passing of photosynthetically active radiation from the red shade net (i.e.red and orange spectra of visible light) enhanced important plant phenomena, such asphotosynthesis and consequently, the biomass.

The pot marigold plants grown under red shading nets produced the highest number of flowers, with the largest flower diameter and the highest total yield compared to the other nets and the open-field plants  (Table 2).

Table 2. Morphological characteristics, yield parameters and total yield

Variety

Treatment

Plant height (cm)

Mass of the whole plant (g)

Number of flowers per  plant

The average mass of the flower (g)

Flower*

diameter (cm)

Flower* fresh yield kg/ha

Flower* dry yield kg/ha

Mix

Pearl

Red

Blue

Control

57.5a

58.2a

57.1a

45.9b

219.6b

250.3a

216.2b

178.9c

 54.18ab

59.77a

44.09c

51.40b

1.006a

1.036a

0.932a

0.936a

5.012b

5.437a

4.983b

4.509c

13537b

15310a

10610c

12117b

2622b

2990a

2048c

2353bc

Domaći

Pearl

Red

Blue

Control

48.3ab

51.2a

48.6ab

38.0c

188.2a

190.6a

189.4a

153.4b

40.88b

46.94a

39.01b

42.57ab

0.88a

0.85a

0.77ab

0.73ab

4.405a

4.550a

4.308a

4.177ab

9032b

10047a

7507c

8102.5bc

1714b

1902a

1467c

1516c

Different superscript letters indicate significant differences within each column (p < 0.05, Duncan’s test)

Line 389: Please clarify this statement; it is not clear which was identified as the most effective.

Samples cultivated under control (open-field) conditions—both domestic and ʹSpringer mixʹ varieties—are grouped on the negative side of both axes. The domestic variety recorded higher total phenolic content (68.16 mg GAE/g d.e.) and flavonoid content (54.84 mg RE/g d.e.) compared to the ʹSpringer mixʹ variety (TPC: 66.84 mg GAE/g d.e.; FC: 39.36 mg RE/g d.e.), with both varieties showing moderate and comparable antioxidant activity (DPPH EC50: 0.079 and 0.080 mg/mL, respectively). These results indicate that the domestic Serbian variety was more effective in accumulating bioactive compounds under control conditions

Line 414: Please clarify this statement.

Among all light treatments, blue nets were most effective in enhancing total extractive matter (TEM) across both varieties. The ʹSpringer mixʹ under blue nets achieved the highest TEM value overall (31.23 ± 0.237 g/100 g d.m.), followed by the domestic variety under blue nets (30.90 ± 0.118 g/100 g d.m.). In comparison, pearl nets yielded the lowest TEM values (22.29 and 24.18 g/100 g d.m., respectively), while red nets and control conditions produced intermediate results.

Line 474 states that other studies have shown net shading consistently increases the phytochemical content of plants; however, the results presented in this manuscript do not support this claim for TPC. This should be further discussed. The results show the highest TPC for the control samples, meaning net shading decreases TPC. This seems to contradict the manuscript's argument, so further discussion or restructuring may be needed.

These results partially align with prior findings in other medicinal and aromatic plants, where spectral shifts induced by netting altered the accumulation of key phytochemicals [16, 18]. However, unlike those studies, total phenolic content in C. officinalis was highest under control (full-spectrum, high-intensity) conditions, suggesting that UV exposure and photoinhibitory stress may be primary drivers of phenolic biosynthesis in this species. Colored nets, while reducing TPC, selectively enhanced other bioactive parameters such as TEM, flavonoids, and radical scavenging activity, demonstrating pathway-specific metabolic regulation.

If there is a considerable lack of information regarding the chemical characterization and profiling of this plant species, perhaps this could be the more direct focus of this manuscript, instead of the net shading.

These results suggest that lighting patterns can be used to modulate the growth and flowering of calendula and to maximize antioxidant activity.

Furthermore, as a comment/question for the authors: If the focus was to compare the shading effect of colored nets against the unshaded open-field cultivation (as mentioned in lines 93-94), why didn't you consider a white shading net as a control for the colored nets (for example)? Because if the unshaded was considered as the control, these plants were exposed to much different conditions (since no shading was applied, regardless of the color). Thus, a white shading net could be used as a control for the shading color.

Pearl nets provide shade and are photo-neutral. In that case, black nets could also be considered as a control for the colored nets. Therefore, plants grown in the open field were used as the control.

Round 2

Reviewer 2 Report

Comments and Suggestions for Authors

Los autores realizaron los cambios recomendados.

Author Response

We hope that we have addressed the minor comments of the reviewers and editors and have significantly improved the article. The English language has been considerably refined, as it was reviewed by a native speaker. We trust that we have met the high standards required by your prestigious journal